# Hyperbolic Variational Graph Auto-Encoder for Next POI Recommendation

## Abstract

Next Point-of-Interest (POI) recommendation has become a crucial task in Location-Based Social Networks (LBSNs), which provide personalized recommendations by predicting the user's next check-in locations. Commonly used models including Recurrent Neural Networks (RNNs) and Graph Convolutional Networks (GCNs) have been widely explored. However, these models face significant challenges, including the difficulty of capturing the hierarchical and tree-like structure of POIs in Euclidean space and the sparsity problem inherent in POI recommendations. To address these challenges, we propose a Hyperbolic Variational Graph Auto-Encoder (HVGAE) for next POI recommendation. Specifically, we utilize a Hyperbolic Graph Convolutional Network (Hyperbolic GCN) to model hierarchical structures and tree-like relationships by converting node embeddings from euclidean space to hyperbolic space. Then we use Variational Graph Auto-Encoder (VGAE) to convert node embeddings to probabilistic distributions, enhancing the capture of deeper latent features and providing a more robust model structure. Furthermore, we combine the Mamba4Rec recommender and Rotary Position Embedding (RoPE) and propose Rotary Position Mamba (RPMamba) to effectively utilize POI embeddings rich in sequential information, which improves the accuracy of the next POI recommendation. Extensive experiments on three public datasets demonstrate the superior performance of the HVGAE model.

## CCS Concepts

• **Information systems → Recommender systems**.

## Keywords

Point-of-interest recommendation, hyperbolic space, variational graph auto-encoder, graph convolutional network, mamba

**ACM Reference Format:**
Anonymous Author(s). 2018. Hyperbolic Variational Graph Auto-Encoder for Next POI Recommendation. In *Proceedings of Make sure to enter the correct conference title from your rights confirmation emai (Conference acronym 'XX).* ACM, New York, NY, USA, 8 pages. https://doi.org/XXXXXXX.XXXXXXX

## 1 Introduction

Next Point-of-Interest (POI) recommendation has become a pivotal task in Location-based Social Networks (LBSNs), aiming to provide

personalized recommendations by predicting users' next check-in locations. Currently, Some works have been accumulated on next POI recommendation, with most methods using deep learning-based techniques to mine user preferences for POIs. Some works utilize Recurrent Neural Networks (RNNs) and their variants, such as Long Short-Term Memory (LSTM) models and Gated Recurrent Unit (GRU), to capture the sequential relationships between user check-ins. However, these methods fail to capture the high-order connectivity between users and POIs. Some other work utilizes Graph Convolutional Networks (GCNs) to address this limitation by capturing the high-order connectivity between users and POIs. However, GCN-based methods often face the issue of over-smoothing and are susceptible to the effects of data sparsity. To mitigate the over-smoothing problem, some works have utilized simplified GCNs and improved versions of GCNs. Additionally, other works have leveraged the parallel processing capabilities of transformers to enhance recommendation efficiency.

Although the above studies have investigated POI recommendations in different aspects, the current methods face major challenges in effectively capturing the complex relationships and hierarchical structures inherent in user check-in behaviors. The first challenge (c1) is that all of the above methods model user preferences in Euclidean space, which makes it difficult to mine the hierarchical relationships and deep feature extraction among POIs. Due to their reliance on Euclidean space, these methods are limited in representing the hierarchical and tree-like structures of POIs. For example, consider a user exploring different categories of POIs such as museums, restaurants, and parks. The user might first visit a general category such as museums and then explore subcategories such as art museums, history museums, and science museums. This exploration pattern forms a hierarchical structure where general categories branch into specific subcategories. Accurately modeling this hierarchy requires a representation that can inherently capture such tree-like relationships, which models based on Euclidean space struggle to achieve. Another significant challenge (c2) is the sparsity in POI recommendations, which severely impacts model performance. Many works have considered various data augmentation techniques to enhance model robustness. However, current research on utilizing embedding transformations into probabilistic distributions for POI recommendation remains inadequate.

To address the above challenges, we propose a model that combines Hyperbolic Graph Convolutional Networks (Hyperbolic GCN) and Variational Graph Auto-Encoders (VGAE) for next POI recommendation, abbreviated as HVGAE. To address challenge (c1), we consider that Hyperbolic GCNs are particularly suitable for accurately modeling hierarchical structures and capturing tree-like relationships, and we propose to use Hyperbolic GCN to transform node representations from Euclidean space to Hyperbolic space to capture the relationships among POIs. To address challenge (c2), we utilize Variational Graph Auto-Encoders (VGAE) to transform node embeddings into probabilistic distributions, which helps to capture

deeper latent features and obtain a more robust model structure through the reconfigured graph structure. VGAE can enhance embeddings with richer structural information, which are then transformed back into embedding representations through hyperbolic graph convolution. Moreover, an advanced mechanism to integrate these enriched embeddings for accurate recommendation is needed in the final recommendation phase. We integrate the Mamba4REC recommender and introduce Rotary Position Embedding (RoPE) to utilize POI embeddings rich in fine-grained information. This positional encoding helps to capture the sequential information of user check-ins, thereby improving the accuracy of the next POI recommendation.

The main contributions of this paper are summarized below:

- We utilize Hyperbolic GCN to capture higher-order interaction information between users and POIs as well as the hierarchical structure of the POIs.
- We propose to transform embeddings into latent variable distributions to achieve deep information capture and enhance the interaction relationships between POIs, thus improving the robustness of the model.
- We propose the RPMamba recommender that adds RoPE to the recommender Mamba4Rec to effectively capture the sequential information of user check-ins, and enhance the model recommendation performance.
- Extensive experiments on three public datasets of different scales validate the performance of our proposed HVGAE model. Furthermore, rational ablation experiments validate the effectiveness of each model component.

## 2 Related Work

### 2.1 Sequential Model for Next POI Recommendation

Sequential models have been widely explored in the field of POI recommendation. Sequential models such as Markov models [8] and RNNs [2] can effectively mine the temporal dependencies within user check-in sequences and then make sequential POI recommendations for users. Markov models are limited by the difficulty of capturing users' long-term preferences due to their no posteriority, while LSTMs [11] and GRUs [16] can capture users' long-term and short-term preferences due to their gated structure. Nevertheless, they are difficult to be trained in parallel, resulting in excessive training costs, and cannot handle dependencies between different lengths. Transformer-based [23, 27, 32] models have been widely utilized by addressing many of the limitations inherent in LSTMs, and their self-attention mechanism allows for parallel computation, which greatly accelerates training and inference time. Recently, inspired by the success of State Space Models (SSMs), Mamba-based methods [4, 15] have emerged to further improve model performance while maintaining inference efficiency.

### 2.2 Graph Neural Networks

The interactions between users and POIs on LBSNs naturally form a bipartite graph, and thus GNNs can effectively capture higher-order connectivity between users and POIs. GCNs utilize the interaction behavior to make POI recommendations. However, GCNs encounter limitations such as over-smoothing, where node embeddings become indistinguishable after multiple layers of convolution. To overcome these limitations, GraphSAGE [5] employs a sampling method to aggregate features from fixed-size neighborhoods, thus enhancing scalability. GATs [24] introduce attention mechanisms to dynamically weigh the importance of neighboring nodes, thus improving the model's expressiveness. NGCF [26] generalizes GNN into the field of collaborative filtering. There are also some simplified GCNs such as lightGCN [9] and SVD-GCN [18] which simplify the feature transformations and nonlinear activations in the original GCN and overcome the over-smoothing problem by various designs. While GCNs in Euclidean space excel in many domains, they have difficulty capturing the hierarchical and tree-like structures inherent in real-world data. This limitation arises from the inability of Euclidean spaces to naturally model such hierarchical structures. To overcome this, some works have explored GCNs in hyperbolic space, such as HGCN [1], HICF [29], HIE [31] and HRCF [30]. Hyperbolic GCNs use the unique property of hyperbolic space in which distances grow exponentially. This property allows hyperbolic GCNs to efficiently embed tree structures and capture hierarchical relationships with fewer dimensions compared to Euclidean space.

### 2.3 Variational Graph Auto-Encoders

VGAEs have emerged as a powerful tool for learning latent representations of graph-structured data, providing a probabilistic approach to encoding graph information. VGAE [13] is proposed to learn interpretable latent representations of undirected graphs using latent variables. Subsequently, ARGA and ARVGA [17] are used as variants of VGAE to obtain robust embeddings through adversarial training. SIG-VAE [6] enhances the flexibility of modeling graph data through a hierarchical variational framework. To solve the noise and sparsity problems, MVGAE [34] is used as a multimodal graph variational auto-encoder to realize the representation of nodes, and the final high-performance recommendation is achieved by fusing the semantic information in multimodality.

## 3 Methodology

### 3.1 Preliminaries

*3.1.1 Next POI Recommendation.* Denote $\mathcal{U} = \{u_1, u_2, ..., u_{|\mathcal{U}|}\}$ as the set of users and $\mathcal{P} = \{p_1, p_2, ..., p_{|\mathcal{P}|}\}$ as the set of POIs. Denote the number of users and POIs as |U| and |P|, respectively. We denote the sequence of check-ins for user $u$ as $S^u = \{s_1^u, s_2^u, ..., s_t^u\}, s_t^u \in P$ is the $t$-th POI checked in by user $u$, and $l_u$ is is the length of $u$'s check-in sequence $S^u$. Given a user's check-in sequence $S^u$, the target of next POI recommendation is to predict the next POI $s_{l_u+1}^u$ where the user $u$ is most likely to check in.

To capture the higher-order connectivity between different POIs, we utilize the check-in history of users to generate a graph $\mathcal{G} = (\mathcal{P}, \mathcal{E})$ to represent the transition relationships among different POIs, where $N = |\mathcal{P}|$ denotes the number of all POIs. $A = (a_{ij}) \in \mathbb{R}^{N \times N}$ represents the adjacency matrix of $\mathcal{G}$, capturing the implicit relationships among POIs, $a_{ij} = 1$ if $(s_i^u, s_j^u) \in \mathcal{E}$ and $a_{ij} = 0$ otherwise. $D \in \mathbb{R}^{N \times N}$ is the degree matrix of $A$. For the edge set $\mathcal{E}$, we process each user sequence and create an edge between each

POI and its $n$-hop neighbors within the sequence, and construct it by the following form:

$$\mathcal{E} = \left\{ \left( s_i^u, s_j^u \right) : u \in \mathcal{U}, |i - j| \leq n, 1 \leq i < j \leq l_u \right\}. \quad (1)$$

3.1.2 *Hyperboloid Manifold.* Let $(p_i^{0,E})_{i \in \mathcal{P}}$ of size $\mathbb{R}^d$ denotes the input POI features, where $^0$ denotes the first layer, the superscript $^E$ denotes the node features lie in a Euclidean space and $^H$ denotes Hyperbolic features. Let $\langle .,. \rangle_{\mathcal{L}} : \mathbb{R}^{d+1} \times \mathbb{R}^{d+1} \to \mathbb{R}$ denotes the Minkowski inner product, $\langle \mathbf{p}, \mathbf{x} \rangle_{\mathcal{L}} := -p_0 x_0 + p_1 x_1 + \ldots + p_d x_d$. We denote $\mathbb{H}_K^d$ as the hyperboloid manifold in $d$ dimensions with constant negative curvature $-1/K (K > 0)$, and $\mathcal{T}_p \mathbb{H}_K^d$ as the (Euclidean) tangent space centered at point p.

$$\mathbb{H}_K^d := \left\{ \mathbf{p} \in \mathbb{R}^{d+1} : \langle \mathbf{p}, \mathbf{p} \rangle_{\mathcal{L}} = -K, p_0 > 0 \right\}, \quad (2)$$

$$\mathcal{T}_p \mathbb{H}_K^d := \left\{ \mathbf{v} \in \mathbb{R}^{d+1} : \langle \mathbf{v}, \mathbf{p} \rangle_{\mathcal{L}} = 0 \right\}. \quad (3)$$

The metric tensor is $g_{\mathcal{L}} = diag[-1, 1, 1, ..., 1]$. Since there is no notion of vector space structure in hyperbolic spaces, it is necessary to implement the derive transformations in hyperbolic models. Specifically, we utilize the exp and log maps to implement Euclidean transformations in the target space $\mathcal{T}_o \mathbb{H}^{d,K}$. For $\mathbf{p} \in \mathbb{H}_K^d$, $\mathbf{x} \in \mathbb{H}_K^d$ and $\mathbf{v} \in \mathcal{T}_p \mathbb{H}_K^d$ such that $\mathbf{v} \neq 0$ and $\mathbf{p} \neq \mathbf{x}$, the exp and log maps of the hyperboloid model are given by:

$$\exp_{\mathbf{p}}^K (\mathbf{v}) = \cosh \left( \frac{\|\mathbf{v}\|_{\mathcal{L}}}{\sqrt{K}} \right) \mathbf{p} + \sqrt{K} \sinh \left( \frac{\|\mathbf{v}\|_{\mathcal{L}}}{\sqrt{K}} \right) \frac{\mathbf{v}}{\|\mathbf{v}\|_{\mathcal{L}}}, \quad (4)$$

$$\log_{\mathbf{p}}^K (\mathbf{x}) = d_{\mathcal{L}}^K (\mathbf{p}, \mathbf{x}) \frac{\mathbf{x} + \frac{1}{K} \langle \mathbf{p}, \mathbf{x} \rangle_{\mathcal{L}} \mathbf{p}}{\left\| \mathbf{x} + \frac{1}{K} \langle \mathbf{p}, \mathbf{x} \rangle_{\mathcal{L}} \mathbf{p} \right\|_{\mathcal{L}}}, \quad (5)$$

where $\|\mathbf{v}\|_{\mathcal{L}} = \sqrt{(\langle \mathbf{v}, \mathbf{v} \rangle_{\mathcal{L}})}$ is the Lorentzian norm of $\mathbf{v}$, the $d_{\mathcal{L}}^K(.,.)$ is the distance between two points $\mathbf{p}, \mathbf{x} \in \mathbb{H}_K^d$ is then:

$$d_{\mathcal{L}}^K (\mathbf{p}, \mathbf{x}) = \sqrt{K} \operatorname{arcosh} \left( -\langle \mathbf{p}, \mathbf{x} \rangle_{\mathcal{L}} / K \right). \quad (6)$$

## 3.2 Hyperbolic Variational Graph Auto-Encoder

3.2.1 *Hyperbolic Initialization Layer.* We first map the embedding of POIs from Euclidean space to hyperbolic space. Let $\mathbf{o} := \{\sqrt{K}, 0, ..., 0\} \in \mathbb{H}_K^d$ denote the north pole (origin) in $\mathbb{H}_K^d$, which we use as a reference point to perform tangent space operations. In particular, an initial hyperbolic node state $\mathbf{p}^{0,H} \in \mathbb{H}^d$ is given by

$$\mathbf{p}^{0,H} = \exp_{\mathbf{o}}^K \left( \left( \mathbf{p}^{0,\mathcal{T}} \right) \right), \quad (7)$$

where $\mathbf{p}^{0,\mathcal{T}} = (0, \mathbf{p}^{0,E})$ and $\mathbf{p}^{0,E} \in \mathbb{R}^d$ is sampled from multivariate Gaussian distribution, the superscript $^{\mathcal{T}}$ denotes that node features lie in a tangent space. So we can interpret $(0, \mathbf{p}^{0,E})$ as a point in $\mathcal{T}_o \mathbb{H}_K^d$ and map it to $\mathbb{H}_K^d$.

3.2.2 *Hyperbolic Message Aggregation.* Considering that feature transformation and non-linear activation in the aggregation process have been verified as unnecessary modules that do not contribute beneficially to model performance [9], we have removed these two components in the hyperbolic space as well. Next, we combine exp and log maps in hyperbolic space to accomplish information aggregation. Specifically, the hyperbolic initial state needs to be projected

to the tangent space via log map. For the Lorentz representation this log map is defined as:

$$\mathbf{p}^{0,\mathcal{T}} = \log_{\mathbf{o}}^K \left( \left( \mathbf{p}^{0,H} \right) \right). \quad (8)$$

Given the POI relationship transformation graph $\mathcal{G}$, we can obtain the neighbors $\mathcal{N}_i$ of the POI $p_i$.

$$p_i^{l+1,\mathcal{T}} = p_i^{l,\mathcal{T}} + \sum_{j \in \mathcal{N}_i} \frac{1}{|\mathcal{N}_i|} p_j^{l,\mathcal{T}}. \quad (9)$$

We apply normalization by degree $|\mathcal{N}_i|$ to ensure that the scale of embeddings does not increase with the number of layers.

To prevent gradient vanishing and over-smoothing, we design the architecture to include skip connections (i.e., skipGCN). Inspired by residual networks [7], we add connections from each layer to the final layer. Finally, we aggregate the information from all layers and obtain the final representation $\mathbf{p}^{sum,\mathcal{T}}$:

$$\mathbf{p}^{sum,\mathcal{T}} = \sum_l \left( \mathbf{p}^{1,\mathcal{T}}, \mathbf{p}^{2,\mathcal{T}}, ..., \mathbf{p}^{l,\mathcal{T}} \right). \quad (10)$$

Then we map $\mathbf{p}^{sum,\mathcal{T}}$ back to the hyperbolic space via the exp map:

$$Z^H = \exp_{\mathbf{o}}^K \left( \mathbf{p}^{sum,\mathcal{T}} \right). \quad (11)$$

3.2.3 *VGAE-driven Graph Transformation.* We convert node representations to latent variable distributions as a way to mine deep information. The encoder consists of two encoding heads:

$$Z^{(1)} = f_{\text{ReLU}} \left( Z^H, A \mid W^{(0)} \right), \quad (12)$$

$$Z_\mu^{(2)} = f_{\text{Linear}} \left( Z^{(1)}, A \mid W_\mu^{(1)} \right), \quad (13)$$

$$Z_\sigma^{(2)} = f_{\text{Linear}} \left( Z^{(1)}, A \mid W_\sigma^{(1)} \right). \quad (14)$$

For **Inference model**, we take a simple inference model parameterized by a two-layer GCN:

$$q(Z \mid Z^H, A) = \prod_{i=1}^N q \left( z_i \mid Z^H, A \right),$$
$$\text{with} \prod_{i=1}^N q \left( z_i \mid Z^H, A \right) = \prod_{i=1}^N \mathcal{N} \left( z_i \mid \mu_{z_i}, \operatorname{diag} \left( \sigma_{z_i}^2 \right) \right), \quad (15)$$

where $q(Z \mid Z^H, A)$ denotes joint distribution of latent variables for all nodes, $q \left( z_i \mid Z^H, A \right)$ denotes the distribution of latent variables for node $z_i$, $\mu_{z_i} = Z_\mu^{(2)}[i,:]$ is the mean vector of a multivariate Gaussian distribution associated within $z_i$, and $\sigma_{z_i}^2 = Z_\sigma^{(2)}[i,:]$ is the corresponding variance vector. The potential representation $z_i$ can be computed using mean and variance:

$$z_i = \mu_i + \sigma \odot \epsilon_i, \quad (16)$$

where $\epsilon_i \sim \mathcal{N}(0, 1)$ is the noise matrix generated by the standard normal distribution, $\odot$ denotes the Hadamard product.

For **Generative model**, we take a generative model parameterized by an inner product between latent variables $z_i$ and $z_j$:

$$p(A \mid Z) = \prod_{i=1}^N \prod_{j=1}^N p \left( a_{ij} \mid z_i, z_j \right),$$
$$\text{with } p \left( a_{ij} = 1 \mid z_i, z_j \right) = \sigma \left( z_i^\top z_j \right), \quad (17)$$

Figure 1: Overall architecture of our proposed VGAE-GT. (a) Item transition graph construction. (b) Variational Graph Auto-Encoder driven graph enhancement. (c) Mbmba4Rec as our backbone for sequence encoding in main recommendation task.

where $a_{ij}$ are the elements of $A$, $\sigma(\cdot)$ is the logistic sigmoid function.

We use Kullback-Leibler Divergence (KLD) to measure the differences between distributions and make the generated distribution to approximate the assumed standard Gaussian distribution as closely as possible. The KLD loss is as follows:

$$\mathcal{L}_{KL} = \text{KL}[q(Z \mid Z^H, A) \| p(Z)]$$
$$= \frac{1}{2} \sum_{i=1}^{N} \left( 1 + \log\left(\sigma_i^2\right) - \mu_{z_i}^2 - \sigma_{z_i}^2 \right). \quad (18)$$

The reconstruction loss first consists of the cross-entropy loss for positive and negative samples, respectively:

$$\mathcal{L}_{\text{reco}} = \mathcal{L}_{\text{pos}} + \mathcal{L}_{\text{neg}}$$
$$= -\sum_{(i,j)\in\text{Pos}} \log(\sigma(z_i \cdot z_j)) \quad (19)$$
$$-\sum_{(i,j)\in\text{Neg}} \log(1 - \sigma(z_i \cdot z_j)).$$

Finally, we formulate a variational lower bound of the input graph log-likelihood as follows:

$$\tilde{\mathcal{L}}_{\text{reco}} = \mathcal{L}_{\text{pos}} + \mathcal{L}_{\text{neg}} - \mathcal{L}_{KL}$$
$$= \sum_{i,j=1}^{N} \mathbb{E}_{z_i, z_j \sim q(.|Z^H, A)} \left[ \log\left( p\left(a_{ij} \mid z_i, z_j\right)\right)\right] \quad (20)$$
$$- KL\left(q\left(z_i \mid Z^H, A\right) \| p\left(z_i\right)\right).$$

### 3.3 RPMamba as Sequence Encoder

After obtaining the updated graph structure through VGAE, we restore the POI representation in the hyperbolic space. We still utilize the hyperbolic GCN to reconstruct the POI representations and ignore weight transformations and nonlinear activations. The POI embeddings (in the form of latent variable distributions) initially obtained from VGAE are mapped back into hyperbolic space. We generate the POI representations through the following process.

$$e_i^{l+1} = e_i^l + \sum_{j \in \mathcal{N}_i} \frac{1}{|\mathcal{N}_i|} e_j^l, \quad \hat{e}^l = \sum_{l=1}^{L} e_i^l. \quad (21)$$

where $L$ is the total number of layers, and $\mathcal{N}_i$ represents the 1-hop neighborhood of POI $p_i$. $e_i^l$ and $e_i^{l+1}$ are the reconstructed embeddings of POI $p_i$ at the $l$-th and $(l+1)$-th layers, respectively. $\hat{e}^l$ is the final reconstructed representation of POI $p_i$. Here, we set a threshold parameter $b$ to determine the presence of edges in the reconstructed graph. If the probability of an edge existing is greater than or equal to $b$, a reconstructed edge is established between the two POIs; otherwise, no edge is formed.

Next, we obtain node representations of the users based on their check-in sequences. Simply adding positional encoding to capture the sequence information of the user's check-in ignores the relative positional information in the user's check-in. Therefore, we add RoPE [21] to model the dependencies between check-in sequences. We denote $\mathbf{m}_i$ as the position encoding vector, and $\mathbf{R}(i)$ as the rotation matrix, which is defined as:

$$\mathbf{R}(i) = \begin{bmatrix} \cos(\theta_i) & -\sin(\theta_i) \\ \sin(\theta_i) & \cos(\theta_i) \end{bmatrix} \quad (22)$$

For each check-in $\hat{e}_{s_i^u}$ and rotation position encoding, the corresponding user representation can be aggregated as:

$$\mathbf{E}_u = \left[ \left(\hat{e}_{s_1^u} + \mathbf{R}(1)\mathbf{m}_1\right), \cdots, \left(\hat{e}_{s_t^u} + \mathbf{R}(t)\mathbf{m}_t\right)\right] \quad (23)$$

Next, we obtain the user check-in sequence with added RoPE, which serves as the final input containing both POI features and positional order. This input is then fed into the Mamba block for training, resulting in the output $\hat{\mathbf{E}}_u$:

$$\hat{\mathbf{E}}_u = Mamba(\mathbf{E}_u). \quad (24)$$

Finally, we utilize the product of the user's representation and the POI's representation to compute the probability of the POI that the user will visit next.

$$\hat{y} = \hat{\mathbf{E}}_{u,t} \cdot \tilde{e}_{s_{t+1}^u}. \quad (25)$$

By incorporating RoPE into the user check-in sequences and utilizing the RPMamba model, we effectively capture both the POI

features and their sequential order. This approach leads to the final recommendation results, enhancing the accuracy and relevance of the recommendations.

## 3.4 Model Training

We utilize the cross-entropy loss function as the loss function for the main POI recommendation task. Additionally, we perform data augmentation using subsequences (i.e., $\{(s_1^u), (s_1^u, s_2^u), ..., (s_1^u, ..., s_{t-1}^u)\}$) of user check-in sequence $S^u$ during the model training process. The loss function for the main task is defined as follows:

$$\mathcal{L}_{\text{main}} = - \sum_{u \in \mathcal{U}} \sum_{1 \le t \le l_u} \log \sigma \left( \hat{E}_{u,t} \cdot \tilde{e}_{s_{t+1}^u} \right) + \log \left( 1 - \sigma \left( \hat{E}_{u,t} \cdot \tilde{e}_{p_t^-} \right) \right),$$
(26)

where $\hat{E}_{u,t}$ represents the embedding of the sequence $(s_1^u, ..., s_t^u)$, and $p_t^- \notin S^u$ is the $t$-th item randomly chosen from the negative samples. To prevent model over-fitting, we also introduce a regularization loss by computing the L2 norm of the model parameters:

$$\mathcal{L}_{\text{reg}} = \|\theta_{\text{en}}\|_2^2 + \|\theta_{\text{de}}\|_2^2 + \|\theta_{\text{recom}}\|_2^2,$$
(27)

where $\theta_{\text{en}}$, $\theta_{\text{de}}$, and $\theta_{\text{recom}}$ are the parameters of the encoder, decoder, and recommender models, respectively. The total loss $\mathcal{L}$ is the weighted sum of the reconstruction loss, the main loss, and the regularization loss:

$$\mathcal{L} = \alpha \tilde{\mathcal{L}}_{\text{reco}} + \beta \mathcal{L}_{\text{main}} + \gamma \mathcal{L}_{\text{reg}}.$$
(28)

where $\alpha$ and $\beta$ are hyperparameters that control the contributions of the reconstruction loss and the main loss, respectively, and $\gamma$ is the regularization factor.

## 4 Experiments

We conduct experiments to evaluate the performance of HVGAE, focusing on the following questions:

- **RQ1:** How does our HVGAE perform as compared to various state-of-the-art recommendation methods?
- **RQ2:** How does the hyperbolic GCN affect model performance in next POI recommendation?
- **RQ3:** How does the VGAE affect model performance?
- **RQ4:** How to demonstrate the effectiveness of RPMamba Recommender?
- **RQ5:** How do different hyperparameters affect the model performance?

## 4.1 Experimental Settings

*4.1.1 Datasets and Evaluation Metrics.* In this work, we use three publicly available datasets: NYC, TKY, and Yelp. NYC and TKY are check-in datasets in New York City and Tokyo collected from Foursquare platform, respectively. Yelp contains user check-in data from Yelp platform. Considering the different characteristics of the respective datasets, we delete the POIs where users check-in less than 5 times in the NYC and TKY datasets, and delete the POIs where users check in less than 20 times in the Yelp dataset. The statistics of the dataset are summarized in Table 1.

We evaluate the performance of the proposed HVGAE model using two widely adopted metrics: Hit Rate (HR) and Normalized Discounted Cumulative Gain (NDCG). These metrics are calculated for top-K recommendations, where K is set to 5 and 10.

**Table 1: Statistics of datasets**

| Dataset | #Users | #POIs | #Interactions | Ave.len. | Density |
|---------|--------|-------|---------------|----------|---------|
| NYC | 1,083 | 9,989 | 179,468 | 165.71 | $1.66e^{-2}$ |
| TKY | 2,293 | 15,177 | 494,807 | 215.79 | $1.42e^{-2}$ |
| Yelp | 36948 | 20950 | 1,299,620 | 35.17 | $1.68e^{-3}$ |

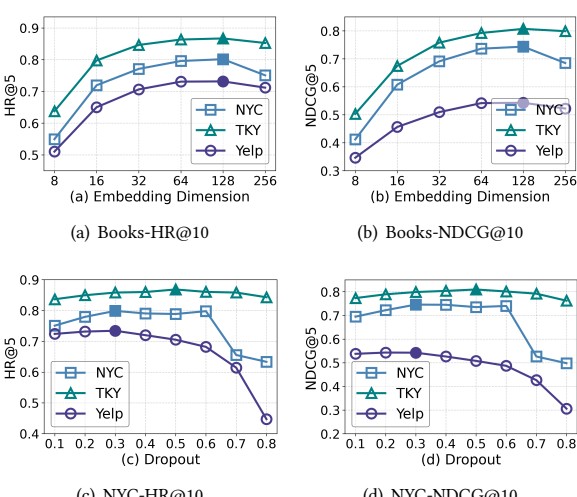

(a) Books-HR@10

(b) Books-NDCG@10

(c) NYC-HR@10

(d) NYC-NDCG@10

**Figure 2: Performance w.r.t. different embedding dimensions and dropouts.**

*4.1.2 Compared Methods.* We compare HVGAE with 10 competitive methods, including attention-based methods: SINE [22], FEARec [3] and TiSASRec [14]; transformer-based methods: CORE [10] and CL4SRec [28]; comparative learning for data enhancement: DouRec [20], MCLRec [19]; diffusion modeling for data augmentation: DiffRec [25]; data enhancement methods combining masks and GNNs: MAERec [33] and AdaMCT [12].

*4.1.3 Experimental Setup.* Our method is implemented in PyTorch and experiments are run on an NVIDIA 4090 GPU. The Adam optimizer is utilized for parameter inference with a learning rate of 1e-2. For the GCN component, we set the number of layers to 2. We apply a regularization coefficient of 1e-6 to improve model generalization. The graph is constructed using a distance parameter of 3. For the parameters of the Mamba block, the SSM state expansion factor is 32, the kernel size for 1D convolution is 4, and the block expansion factor for linear projections is 2.

## 4.2 Overall Performance (RQ1)

We compare the proposed HVGAE with current state-of-the-art methods and summarize the experimental results in Table 2. First, it can be concluded from the Table 2 that HVGAE consistently outperforms the other compared methods on the three publicly available datasets. Specifically, we compare HVGAE with several attention-based sequence recommendation methods, including SINE [22], FEARec [3] and TiSASRec [14]. The performance of such methods is lower overall than that of HVGAE because they can only obtain

Table 2: Model performance.

| Datasets | NYC | | | | TKY | | | | Yelp | | | |
|---|---|---|---|---|---|---|---|---|---|---|---|---|
| Metric | HR@5 | HR@10 | ND@5 | ND@10 | HR@5 | HR@10 | ND@5 | ND@10 | HR@5 | HR@10 | ND@5 | ND@10 |
| SINE | 0.6842 | 0.7442 | 0.6224 | 0.6416 | 0.7745 | 0.8382 | 0.6976 | 0.7184 | 0.5915 | 0.7714 | 0.4156 | 0.4739 |
| CORE | 0.7461 | 0.8172 | 0.6911 | 0.7142 | 0.8439 | 0.8901 | 0.7699 | 0.7850 | 0.6984 | 0.8446 | 0.5087 | 0.5563 |
| CL4SRec | 0.5891 | 0.6704 | 0.5115 | 0.5378 | 0.7418 | 0.8125 | 0.6465 | 0.6693 | 0.6531 | 0.8206 | 0.4623 | 0.5168 |
| DuoRec | 0.7599 | 0.8061 | 0.6990 | 0.7139 | 0.8548 | 0.9001 | 0.7849 | 0.7994 | 0.6995 | 0.8518 | 0.5089 | 0.5585 |
| FEARec | 0.7747 | 0.8135 | 0.7110 | 0.7235 | 0.8522 | 0.8945 | 0.7811 | 0.7949 | 0.7078 | 0.8512 | 0.5243 | 0.5710 |
| MAERec | 0.6851 | 0.7830 | 0.5985 | 0.6300 | 0.7366 | 0.8173 | 0.6252 | 0.6516 | 0.5864 | 0.7761 | 0.4061 | 0.4677 |
| TiSASRec | 0.7729 | 0.8236 | 0.7141 | 0.7304 | 0.8443 | 0.8892 | 0.7808 | 0.7955 | 0.7021 | 0.8510 | 0.5138 | 0.5623 |
| AdaMCT | 0.7581 | 0.8144 | 0.7017 | 0.7199 | 0.8474 | 0.8905 | 0.7797 | 0.7935 | 0.7074 | 0.8514 | 0.5210 | 0.5678 |
| MCLRec | 0.7802 | 0.8319 | 0.7291 | 0.7457 | 0.8447 | 0.8823 | 0.7711 | 0.7832 | 0.6087 | 0.7212 | 0.4549 | 0.4915 |
| DiffRec | 0.7636 | 0.8061 | 0.7060 | 0.7198 | 0.8099 | 0.8513 | 0.7276 | 0.7408 | 0.6087 | 0.7212 | 0.4549 | 0.4915 |
| **HVGAE** | **0.8033** | **0.8421** | **0.7458** | **0.7580** | **0.8683** | **0.9071** | **0.8094** | **0.8221** | **0.7342** | **0.8775** | **0.5423** | **0.5890** |
| *Improv.* | 2.96% | 1.23% | 2.29% | 1.65% | 1.58% | 0.78% | 3.12% | 2.84% | 3.73% | 3.01% | 3.44% | 3.15% |

Table 3: Ablation study with key modules.

| Datasets | NYC | | | | TKY | | | | Yelp | | | |
|---|---|---|---|---|---|---|---|---|---|---|---|---|
| Metric | HR@5 | HR@10 | ND@5 | ND@10 | HR@5 | HR@10 | ND@5 | ND@10 | HR@5 | HR@10 | ND@5 | ND@10 |
| HVGAE-$h1$ | 0.7876 | 0.8236 | 0.7398 | 0.7516 | 0.8648 | 0.9062 | 0.8044 | 0.8179 | 0.7283 | 0.8768 | 0.5389 | 0.5873 |
| HVGAE-$h2$ | 0.7682 | 0.8190 | 0.7107 | 0.7271 | 0.8539 | 0.8927 | 0.7922 | 0.8048 | 0.7236 | 0.8741 | 0.5325 | 0.5815 |
| HVGAE-$h$ | 0.7553 | 0.8199 | 0.6994 | 0.7205 | 0.8434 | 0.8914 | 0.7728 | 0.7883 | 0.7105 | 0.8602 | 0.5201 | 0.5688 |
| HVGAE-$v$ | 0.7747 | 0.8283 | 0.7182 | 0.7357 | 0.8474 | 0.8927 | 0.7799 | 0.7945 | 0.7232 | 0.8724 | 0.5296 | 0.5782 |
| HVGAE-$m$ | 0.7839 | 0.8403 | 0.7279 | 0.7461 | 0.8565 | 0.9014 | 0.7960 | 0.8080 | 0.7201 | 0.8689 | 0.5256 | 0.5741 |
| HVGAE-$p$ | 0.7775 | 0.8218 | 0.7185 | 0.7328 | 0.8583 | 0.8949 | 0.7974 | 0.8126 | 0.7210 | 0.8691 | 0.5275 | 0.5757 |
| HVGAE-$rp$ | 0.7821 | 0.8319 | 0.7320 | 0.7486 | 0.8657 | 0.8993 | 0.8085 | 0.8187 | 0.7241 | 0.8739 | 0.5323 | 0.5811 |
| **HVGAE** | **0.8033** | **0.8421** | **0.7458** | **0.7580** | **0.8683** | **0.9071** | **0.8094** | **0.8221** | **0.7342** | **0.8775** | **0.5423** | **0.5890** |

user preferences from a limited amount of sparse data. secondly, we compare HVGAE with transformer-based models (i.e., CORE [10], CL4SRec [28]) that are currently widely used in sequence recommendation. The performance of such methods is also overall lower than HVGAE due to the lack of data augmentation phase and the fact that they are not as fast as mamba inference. Considering that we perform graph-structured data augmentation using VGAE, we also compare HVGAE with many data-augmented recommendation methods. For example, the recommended method MAERec [33], which utilizes masks for data enhancement combined with GCN, and so on. As a result of the comparison experiments, HVGAE is also outperforms the other comparison methods due to its ability to capture higher-order information in hyperbolic space, capture higher-order information between POIs using latent space, and propose the current state-of-the-art recommender RPMamba.

## 4.3 Ablation Study

*4.3.1 Hyperbolic GCN (RQ2).* To validate the effectiveness of capturing hierarchical relationships among POIs in hyperbolic space, we have conducted ablation experiments, with results summarized in Table 3. We propose several model variants, HVGAE-$h$ denotes the elimination of the process of capturing the user's neighbor information in the hyperbolic GCN, with all message passing and graph structure reconstruction are implemented by LightGCN. HVGAE-$h1$ removes hyperbolic GCN during the message passing phase, using

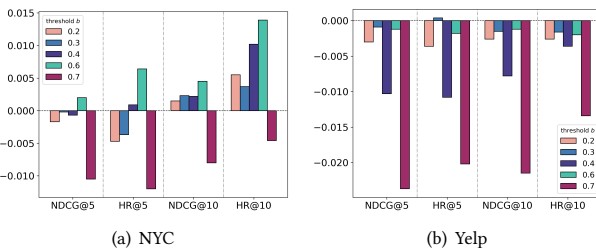

(a) NYC                    (b) Yelp

Figure 3: Performance w.r.t. different threshold values $b$.

traditional GCN for capturing user node information via the item transition graph, while utilizing hyperbolic GCN for graph structure reconstruction. HVGAE-$h2$ removes hyperbolic GCN during the graph structure reconstruction phase, using hyperbolic GCN to capture user node information in the item transition graph and using lightweight LightGCN for graph structure reconstruction. From the experimental results, it can be concluded that there is a significant degradation of the performance of HVGAE-h, demonstrating the critical role of hyperbolic GCN in both message passing and graph structure reconstruction. Hyperbolic GCNs can effectively capture the hierarchical relationships among POIs in hyperbolic space. The superior performance of HVGAE-$h1$ and HVGAE-$h2$ over HVGAE-$h$ indicates the unique contributions of hyperbolic

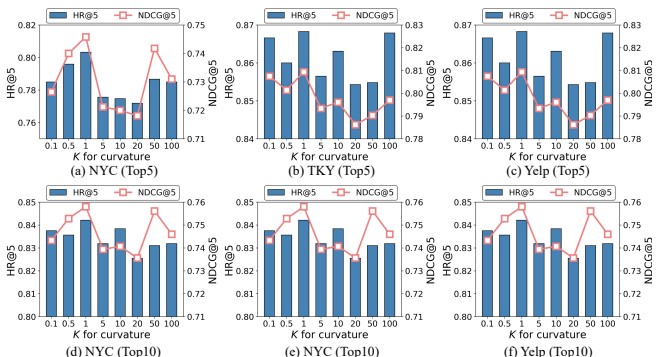

**Figure 4: Performance w.r.t. different $K$ values for curvature.**

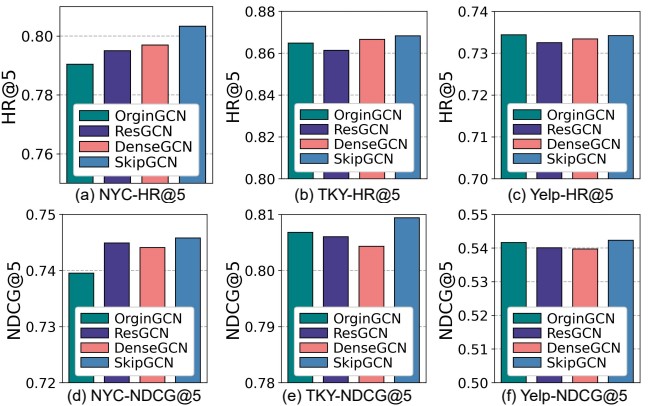

**Figure 5: Performance w.r.t. different GCN architectures.**

GCN in different stages. Additionally, the better performance of HVGAE-$h1$ than HVGAE-$h2$ then suggests that hyperbolic GCNs are slightly more influential in the graph reconstruction phase than in the message passing phase.

As mentioned in Subsection 3.2.2, To prevent oversmoothing, we propose skipGCN. This not only prevents over-smoothing of the HVGAE model but also effectively improves the model performance. We compare our proposed skipGCN with graph convolution models of some other architectures and summarize the comparison results in Figure 5. It can be concluded that the performance of our model outperforms other architectures (i.e., originGCN, ResGCN, and DenseGCN) on all three publicly available datasets.

*4.3.2 Variational Graph Auto-Encoder (RQ3).* VGAE uses variational inference to convert node embeddings into probabilities, and the conversion can capture deeper latent features, which helps to improve the robustness and generalization of the model. To validate the effectiveness of the VGAE module, we remove this module and propose its corresponding variant: HVGAE-$v$. The results are summarized in Table 3. The degradation of the performance of HVGAE-$v$ indicates that without the probability conversion and deep feature extraction facilitated by VGAE, the complex features between POIs are difficult to be mined, which leads to the degradation of the model's performance.

*4.3.3 RPMamba Recommender (RQ4).* To validate the effectiveness of our proposed RPMamba, we replace RPMamba with SASRec, an excellent Transformer-based model, and then perform ablation experiments and propose a variant HVGAE-$m$. The results are summarized in Table 3. It can be concluded that HVGAE-$m$ performs significantly weaker than the original HVGAE model using RP-Mamba as the recommender. This is because RPMamba is based on a selective state-space model, which exhibits higher efficiency and performance when dealing with long sequences. In addition, Mamba's architecture removes the traditional attention mechanism and multi-layer perceptron block, which makes it simpler than the Transformer-based model architecture.

*4.3.4 Rotary Position Embedding.* To investigate the importance of RoPE, we conduct ablation experiments. We remove the RoPE and obtain its corresponding variant: HVGAE-$p$. Additionally, to verify the unique ability of RoPE to capture relative positional relationships within sequences compared to ordinary positional encoding, we replace the RoPE with ordinary positional encoding and propose its variant: HVGAE-$rp$. Experimental results for HVGAE-$p$ and HVGAE-$rp$ have been summarized in Table 3. The performance degradation of these variants demonstrates that the ability of RoPE to capture relative positions significantly enhances the performance of our model in sequential POI recommendations.

## 4.4 Hyper-parameters (RQ5)

*4.4.1 Embedding Dimension.* The embedding dimension affects the performance of the model. To explore the embedding dimensions most adapted to our model, we provide several dimension candidates (i.e., 8, 16, 32, 64, 128, 256) and conduct experiments on all datasets for comparison. Specifically, we choose HR@10 and NDCG@10 as the measures, and the comparison results are summarized in subfigures 2(a) and 2(b) of Figure 2. As can be seen from the figures, the model's performance reaches its optimal value when dimension = 128, and the performance of the subsequent models starts to decrease. Therefore, we choose to embedding dimension = 128 in all datasets used in this paper.

*4.4.2 Dropout.* We use dropout to prevent model overfitting. We give the corresponding candidates (i.e., 0.1, 0.2, 0.3, 0.4, 0.5, 0.6, 0.7, 0.8), choose HR@10 and NDCG@10 as the measures, and then conduct experiments on three datasets. The comparison results are summarized in subfigures 2(c) and 2(d) of Figure 2. By observing the experimental results of the model on different datasets, we finally determine dropout values of 0.3, 0.5 and 0.3 on the NYC, TKY and Yelp datasets, respectively.

*4.4.3 Threshold b.* After the VGAE module, we need to reconstruct the POI relationship graph, and the threshold set here will affect the structure of the reconstructed graph and thus the performance of the model. To explore the impact of the threshold we have conducted thorough experiments and summarized the results of the experiments on the NYC and Yelp datasets as examples in Figure 3. We use the threshold $b$ = 0.5 as a standard to observe the effect of other thresholds and can conclude that the best performance is achieved when the threshold $b$ = 0.6 in the NYC dataset. In the Yelp dataset performance is best when threshold $b$ = 0.5. By way of a

similar comparison, it is concluded that the best performance is achieved in the TKY dataset when the threshold $b$ = 0.5.

*4.4.4 K for curvature.* To investigate the effect of curvature on model performance, we conduct experiments and summarize the results in Figure 4. By exploring the results in Figure 4, we can conclude that the model always achieves optimal performance in the NYC and Yelp datasets when $K$ = 1. The model achieves the optimal value in the vast majority of cases in the TKY dataset and only achieves a sub-optimal value for HR@10. However, it does not affect the final conclusion, i.e., $K$ = 1 was chosen for all three datasets.

## 5  Conclusion

This paper explores a graph augmentation method based on distribution transformation, utilizing VGAE to enhance graph structure information. The embedding is converted into latent variables using VGAE, and then the hidden information is mined and then reduced to the representation of embedding using GCN. Finally, the embeddings are combined with the RPMamba recommender to enhance the sequential recommendation system. We conduct extensive experiments on three real-world datasets and demonstrate that our HVGAE outperforms state-of-the-art baselines. In future work, we plan to design more adaptive graph structure augmentation criteria to further improve the model's adaptability.

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
