# OpenReview forum: "Hyperbolic Variational Graph Auto-Encoder for Next POI Recommendation"
_ACM.org/TheWebConf/2025/Conference — WWW 2025 Poster_

### Official Review · Reviewer_VE33 · 2024-11-01

**Novelty:** 5
**Technical Quality:** 3

**Review:**

This paper addresses two challenges in the task of next POI (Point of Interest) recommendation: 1) modeling user preferences in Euclidean space, which fails to capture the hierarchical relationships between POI categories; and 2) the insufficient research on using embedding transformations into probabilistic distributions to tackle data sparsity. The authors propose a novel model, HVGAE, which employs Hyperbolic GCN to map node representations from Euclidean space to Hyperbolic space, effectively capturing node relationships. Additionally, a Variational Graph AutoEncoder (VGAE) is utilized to constrain the latent variable distribution. The introduction of RPMamba further enhances the model's ability to capture sequential relationships. Performance comparisons are conducted using three datasets, providing valuable insights into the model's effectiveness.



Pros

- The paper introduces a novel technology stack for next POI recommendation, incorporating Hyperbolic GCN, VGAE, and Mambda.
- The paper is well-organized
- The experimental section is comprehensive conducted.

Cons

- The paper lacks a clear explanation of why capturing node relationships in Hyperbolic space is superior to doing so in Euclidean space. An intuitive understanding of this advantage would greatly benefit the reader. Additionally, relevant citations supporting this claim are missing, which could enhance the robustness of the argument.
- The statement in lines 102 to 104, "However, current research on utilizing embedding transformations into probabilistic distributions for POI recommendation remains inadequate," is somewhat confusing. The phrase "utilizing embedding transformations into probabilistic distributions" lacks clarity regarding its intended purpose and the mechanism by which this approach effectively addresses the sparsity issue. The description of this challenge is insufficient and does not convincingly articulate its significance or relevance to the proposed methodology.

- The motivation for introducing the three new technology stacks is not sufficiently explained in the manuscript. While these technologies are indeed prominent in recent research, a clearer articulation of their relevance to the study and how they contribute to the proposed approach would enhance the overall clarity and impact of the work.
- The most recent baseline models should be employed for comparison, as indicated in references [1-4]. Additionally, the baseline models used in the experiments predominantly focus on sequential recommendation rather than being specifically designed for next POI recommendation.  It is crucial to include comparisons with additional relevant studies in this field, such as those presented in references [5-7], to ensure a more thorough evaluation of the proposed methodology.
- The experimental results indicate that the baseline model achieves an HR@5 score exceeding 0.7. However, for datasets from Foursquare, such as NYC and TKY, other literature does not report such high performance. It is essential to address the reasons behind this performance discrepancy. Was there any special processing applied to the dataset that could explain this variance in results?
- The code for the proposed methodology is not provided.



[1] Xu, Y., Cong, G., Zhu, L., & Cui, L. (2024, May). MMPOI: A Multi-Modal Content-Aware Framework for POI Recommendations. In *Proceedings of the ACM on Web Conference 2024* (pp. 3454-3463).

[2] Zhou, H., Jia, Z., Zhu, H., & Zhang, Z. (2024, July). Cllp: Contrastive learning framework based on latent preferences for next poi recommendation. In *Proceedings of the 47th International ACM SIGIR Conference on Research and Development in Information Retrieval* (pp. 1473-1482).

[3] Li, P., de Rijke, M., Xue, H., Ao, S., Song, Y., & Salim, F. D. (2024, July). Large language models for next point-of-interest recommendation. In *Proceedings of the 47th International ACM SIGIR Conference on Research and Development in Information Retrieval* (pp. 1463-1472).

[4] Huang, T., Pan, X., Cai, X., Zhang, Y., & Yuan, X. (2024, March). Learning Time Slot Preferences via Mobility Tree for Next POI Recommendation. In *Proceedings of the AAAI Conference on Artificial Intelligence* (Vol. 38, No. 8, pp. 8535-8543).

[5] Yan, X., Song, T., Jiao, Y., He, J., Wang, J., Li, R., & Chu, W. (2023, July). Spatio-temporal hypergraph learning for next POI recommendation. In *Proceedings of the 46th international ACM SIGIR conference on research and development in information retrieval* (pp. 403-412).

[6] Yin, F., Liu, Y., Shen, Z., Chen, L., Shang, S., & Han, P. (2023, June). Next POI recommendation with dynamic graph and explicit dependency. In *Proceedings of the AAAI Conference on Artificial Intelligence* (Vol. 37, No. 4, pp. 4827-4834).

[7]  Wang, Z., Zhu, Y., Wang, C., Ma, W., Li, B., & Yu, J. (2023, July). Adaptive Graph Representation Learning for Next POI Recommendation. In *Proceedings of the 46th International ACM SIGIR Conference on Research and Development in Information Retrieval* (pp. 393-402).

**Questions:**

Please refer to the aforementioned weaknesses for further details.

**Reviewer Confidence:**

4: The reviewer is certain that the evaluation is correct and very familiar with the relevant literature

**Scope:**

3: The work is somewhat relevant to the Web and to the track, and is of narrow interest to a sub-community

---

### Official Review · Reviewer_7QQq · 2024-12-01

**Novelty:** 5
**Technical Quality:** 4

**Review:**

This paper proposes a POI-recommendation method (HVGAE) in hyperbolic space to address two challenges: the difficulty of mining the hierarchical relationships and deep feature extraction among POIs and the data sparsity in the traditional POI-recommendation method. The authors utilized Hyperbolic GCN to model the structure of the points and combined Mamba4Rec recommender and Rotary Position Embedding to utilize POI embedding effectively. And then the authors compare their method (HVGAE) with some SOTA methods in three datasets: NYC, TKY and Yelp to verify the performance of their method. In top5 hit rate, HVGAE perform great. And it's a very interesting attempt in POI recommendation.

Strengths:

1. This paper innovatively maps the POI embedding into hyperbolic space to address some difficulties faced in traditional Euclidean space.
2. The experiments demonstrate the effectiveness of the proposed method and provide some directions for the next POI recommendation.
3. This paper is well-organized and the descriptions of experiments are clear.

Weaknesses:

1. Lack of explanation about some critical operations in the detailed method description.
2. The generalizability of the proposed method needs to be further evaluated.

**Questions:**

The TKY dataset and NYC dataset perform better than the Yelp dataset. As we know, Yelp is a huge dataset that may include the POI of TKY and NYC. And is the performance affected by the number of POIs? Should the authors include a substantial number of large datasets to verify their method has generalization ability in typical POI recommendations?

There is a careless mistake that needs attention: there's double "is" in Line 219:  ... and $u_s$ is is the length of 𝑢’s check-in sequence $S_u$ ...

**Reviewer Confidence:**

3: The reviewer is confident but not certain that the evaluation is correct

**Scope:**

4: The work is relevant to the Web and to the track, and is of broad interest to the community

---

### Official Review · Reviewer_Pc35 · 2024-12-01

**Novelty:** 4
**Technical Quality:** 5

**Review:**

It proposes a hyperbolic VGAE for next POI recommendation, and extensive experiments have demonstrated its superiority. Overall, this paper is organized and written clearly.

**Questions:**

1. The paper lacks an introduction to the Mamba model.
2. The existing mamba-based methods are not compared.
3. Insufficient experimental analysis，such as the time cost.

**Reviewer Confidence:**

3: The reviewer is confident but not certain that the evaluation is correct

**Scope:**

4: The work is relevant to the Web and to the track, and is of broad interest to the community

---

### Official Review · Reviewer_cP14 · 2024-12-02

**Novelty:** 4
**Technical Quality:** 4

**Review:**

Pros:
P1: The method effectively captures the hierarchical and tree-like relationships of POIs using Hyperbolic GCNs, which outperform traditional Euclidean approaches in modeling complex structures and relationships.

P2: By transforming embeddings into probabilistic distributions, VGAE enhances latent feature mining and addresses data sparsity, improving the model's robustness and representation of high-order connectivity.

Cons:
C1: The construction of edges by considering POIs in each sequence and their n-hop neighbors results in equal weights for 1-hop and 2-hop connections. This is somewhat unreasonable, as the last visited POI should have a greater influence on the current POI than the one visited two steps earlier.

C2: In Equation (25), user embeddings need to combine with target POI embeddings to compute the probability of a user visiting a specific POI. While this design works during training, it poses challenges during inference. Since the next POI is unknown, the user embedding must combine with embeddings of all POIs to compute predictions, which is computationally expensive. The issue worsens when the number of POIs is large, leading to significantly increased inference time.

C3: The title includes "for Next POI Recommendation," so the experiments should consider baselines specifically tailored for this task. The authors could incorporate state-of-the-art baselines, such as sequence-based models [1] and graph-based models [2], to strengthen the evaluation.

C4: In POI recommendation, there exist methods that organize relationships between POIs using tree-like structures, but these are based on the temporal order of POI visits. The authors could include such methods in the related works section [1].


[1] Learning Time Slot Preferences via Mobility Tree for Next POI Recommendation
[2] Spatio-Temporal Hypergraph Learning for Next POI Recommendation

**Questions:**

Q1: In line 224, the authors state, "To capture the higher-order connectivity between different POIs, we utilize the check-in history of users to generate a graph." Does this mean that the graph is generated using the check-in history of all users prior to the current sequence or only the check-in history of the user for the current sequence? Either way, this raises a concern: each sequence would require a separately constructed graph because the start and end times differ across sequences. If this is the case, the graph construction process would be extremely time-consuming and memory-intensive. Could the authors clarify how the graph is actually constructed in practice?

**Reviewer Confidence:**

4: The reviewer is certain that the evaluation is correct and very familiar with the relevant literature

**Scope:**

4: The work is relevant to the Web and to the track, and is of broad interest to the community